# It's COMPASlicated: The Messy Relationship between RAI Datasets and Algorithmic Fairness Benchmarks

**Michelle Bao**
baom@stanford.edu

**Angela Zhou**
angela-zhou@berkeley.edu

**Samantha Zottola**
szottola@prainc.com

**Brian Brubach**[*]
bb100@wellesley.edu

**Sarah Desmarais**[*]
sdesmarais@prainc.com

**Aaron Horowitz**[*]
ahorowitz@aclu.org

**Kristian Lum**[*]
kristianl@twitter.com

**Suresh Venkatasubramanian**[*]
suresh_venkatasubramanian@brown.edu

## Abstract

Risk assessment instrument (RAI) datasets, particularly ProPublica's COMPAS dataset, are commonly used in algorithmic fairness papers due to benchmarking practices of comparing algorithms on datasets used in prior work. In many cases, this data is used as a benchmark to demonstrate good performance without accounting for the complexities of criminal justice (CJ) processes. However, we show that pretrial RAI datasets can contain numerous measurement biases and errors, and due to disparities in discretion and deployment, algorithmic fairness applied to RAI datasets is limited in making claims about real-world outcomes. These reasons make the datasets a poor fit for benchmarking under assumptions of ground truth and real-world impact. Furthermore, conventional practices of simply replicating previous data experiments may implicitly inherit or edify normative positions without explicitly interrogating value-laden assumptions. Without context of how interdisciplinary fields have engaged in CJ research and context of how RAIs operate upstream and downstream, algorithmic fairness practices are misaligned for meaningful contribution in the context of CJ, and would benefit from transparent engagement with normative considerations and values related to fairness, justice, and equality. These factors prompt questions about whether benchmarks for intrinsically socio-technical systems like the CJ system can exist in a beneficial and ethical way.

## 1 Introduction

News stories like ProPublica's Machine Bias [1] forced RAIs (and the COMPAS dataset) into the spotlight of academic and public scrutiny. Since then, discussions of algorithmic fairness within the CS community have relied heavily on the COMPAS dataset as a benchmarking tool both for research on how to build RAIs and as a general source of data for evaluating "fairness" of algorithms. Benchmarking practices include assessing algorithmic improvements on RAI datasets or studying specific data-bias issues but incorrectly assuming RAI datasets provide a ground-truth and thus adding additional data corruption.

This technical focus on CJ data as a vehicle for benchmarking ignores the contextual grounding that is critical to working with RAI datasets and the interdisciplinary work that has shaped RAI use in the CJ system. Every step of the RAI pipeline, from data generation to model prediction to evaluation metrics to deployment, contains decisions that contribute to the fairness of outcomes and the end goals of fairness, justice, and equality. These decisions are guided by principles which draw from the

---

[*]These authors are listed in alphabetical order

35th Conference on Neural Information Processing Systems (NeurIPS 2021) Track on Datasets and Benchmarks.

disciplines of criminology, psychology, science, technology, and society, sociology, philosophy, law, and ethics. A narrow technical focus on dataset bias and benchmarking, while more efficient and consistent, fails to recognize the degree to which datasets are necessarily situated within a broader socio-technical frame [2] of racial injustice and theories of incarceration, resulting in misleading and misguided conclusions about the value of fairness-enhancing tools and their role in CJ.

In this paper we, an interdisciplinary group of researchers and practitioners, provide insight into the complexities of the CJ context embedded in RAI fairness datasets and research and reveal the hidden dimensions of justice and equality that the technical discourse does not acknowledge. We first overview the biases and errors in RAI datasets to demonstrate how compounding distributional errors hinder real-world relevance and generalization. Then, we interrogate claims around real-world impact of RAI algorithmic fairness by examining a) the disconnect between fairness metrics and real-world fairness and b) the implicit values of utilizing RAI datasets on higher values of justice and power. Then, we provide a multidisciplinary lens into methodological standards and guidelines used in disciplines that have historically studied RAIs, and contrast these standards with algorithmic fairness publication practices to understand the differences and incompatibility between the two. We end with a call to arms that summarizes best and harmful practices around using RAI datasets.

## 2   Background and Related Work

Criminology, psychology, and other disciplines have studied RAIs for decades and developed them to improve upon unstructured, human decision-making [3]. There are hundreds of instruments that vary in the amount of structure they provide, who can implement them, the information they query, the amount of information they generate, the decision they are intended to inform, and their level of transparency [4]. RAIs are developed using a combination of theory and analysis to guide the selection of factors associated with particular outcomes (e.g., engagement in violence, rearrest, reconviction, failure to appear, etc.) [3, 5]. Factors vary by tool and may include socio-demographic characteristics like age or gender, as well as other information such as criminal record, education status, or employment history [6]. Factors are scored based on the strength of their association with the outcome they are intended to predict and summed to produce a risk score [7, 8, 9]. This score corresponds to an estimated likelihood of the outcome of interest. While not all RAIs are exclusively algorithmic, we focus on those that are as they have been at the center of the algorithmic fairness discussion.

RAI scores and predictions are currently used across the country to aid decisions in areas like pretrial release conditions, bail determinations, sentencing decisions, parole supervision, probation eligibility, and more [4, 10, 11]. They have been developed by a variety of actors, including for-profit companies, non-profits, researchers, and academics, and vary in their level of transparency. Thus far, there is little conclusive research on how RAIs in general impact key outcomes like incarceration rates, diversion, racial disparities, crime, etc. in the long term [12], driven by vast differences in deployment methods as well as in how the measures are evaluated for fairness and justice.[2] Proponents of RAIs believe the instruments can increase consistency and transparency in decision-making and reduce detention rates [13, 14]. Critics have raised concerns that RAIs themselves are not racially neutral and will contribute to racist decision-making, lack mechanisms of transparency for defendants or accountability for decision-makers, and overstate their applicability in the CJ context [15, 16, 17, 18]. Recognizing limitations of fairness metrics has shifted the focus from unfair RAIs to unfair RAI outcomes in real-world scenarios. "Determining whether a risk tool is racially biased is probably redundant [...] The important question is whether the use of actuarial risk assessment tools results in more disparate outcomes than the status quo, or other viable alternatives" [12].

Within the broader discussions around algorithmic fairness, our work is situated within a growing body of scholarship that emphasizes the importance of understanding training data provenance and context [19, 20] as a part of the model building process. In practice, benchmarking, particularly to demonstrate real-world applicability, complicates this emphasis. Criteria for benchmark evaluation exist in bioinformatics, with the most relevant being relevance and solvability [21]. ML studies have mainly explored the former through "benchmark misalignment" which arises with systematically unstable labels [22, 23], but misalignment in the context of socio-technical systems is rarely discussed.

---

[2]The ongoing debate about the interpretation of COMPAS that ProPublica published is one example of the latter.

# 3 Data Biases and Errors in Pretrial RAIs

Fairness in machine learning papers that develop novel methodological contributions evaluate performance on data assuming that covariates, protected attribute(s), and outcome variables follow a fixed joint distribution: $(X, A, Y) \sim D$. We overview the statistical bias in pretrial RAIs via technical issues in the label $Y$, protected attribute $A$, covariates $X$, and distribution. For each issue, we introduce evidence from CJ and recent work in CS that tries to address it. In Appendix A.1 we include more technical details on some of the methodological proposals. We do not suggest technology can "solve" data bias: evidence from CJ suggests measurement error is both inherent but difficult to calibrate since there is no ground-truth. Hence, for the purposes of generic benchmarking of algorithmic interventions, performance evaluation using RAI data is statistically biased. Issues of measurement relate broadly to both measurement theory in the social sciences [24] and measurement error in econometrics.

We focus the critique on *pretrial* RAIs since the widely used COMPAS dataset is one, though there are other RAIs used in the context of parole, sentencing, community supervision, or hospitals [25]. Our argument does extend to RAIs used in these other contexts when analogous data bias issues arise. For example, probation/parole models are subject to similar label bias issues, such as heightened parole officer involvement or left-censoring when people go back to jail/prison for technical violations and hence censor the possibility of re-arrest [26, 27]. On the other hand RAIs for contexts besides pretrial are not subject to the same FTA label bias and may be more narrowly bracketed in clinical or forensic psychology settings.

**Bias in $Y$.** We first consider the outcome value, $Y$. Typically in pretrial RAIs, the outcome variable chosen is recidivism during the pretrial period or failure to appear (FTA) for court appointments. Both of these primary outcomes exhibit *label bias* due to construct invalidity [24] and measurement bias. On construct validity, legal cases accept that preventing pretrial flight and violent crime are among the only concerns that justify pretrial detention.[3] Whether or not the available substitutes of FTA and re-arrest are aligned with these constructs remains an open question [16, 28, 29].

On measurement bias, due to the fundamental unobservability of crime itself, outcome measures of recidivism that rely on data on "re-arrest" might miss the desired target of "re-offense." Re-arrest as a substitute for re-offense is concerning because measurement bias is correlated with $A$, due to spatially differentiated policing practices and/or other biases in arrest. Fogliato et al. [30] finds that although violent crime arrest rates are proposed to be "unbiased" measures of offense, violent arrest rates still vary in attributes of the offense, including race. One strategy other fields have used to try to overcome this issue is to include self-reported re-offense [31], which increases accuracy of measured violence and crime [32, 33]. Similarly, although models that predict FTA are often described as predicting "flight risk", FTA generally includes all "nonappearance", not only cases of "flight" (absconding from the legal system). As reported via surveys, people do not appear because of scheduling, work, and transportation difficulty, which are correlated with race and class inequality [34]. Thus, conflating "flight" and "non-appearance" introduces measurement error correlated with race and class. Perhaps worse, this creates a prediction outcome less tailored for providing interventions targeted at the root causes of non-appearance, such as free childcare at court, transportation benefits, and text message reminders [35]. Measuring FTA as a binary outcome ignores the many occasions on which people have appeared for court appointments; the distribution of time/schedule precarity suggests measurement error is again correlated with $A$.

Fairness in machine learning has in turn studied questions related to label bias. Fogliato et al. [36] suggests sensitivity analysis, probing how classification results change for different possible amounts of label bias. Wang et al. [37] studies classification with group-dependent label noise: ensuring fairness under biased label noise can harm both accuracy and fairness (evaluated with true labels). However, calibrating noise models may be particularly difficult with CJ data due to the generic lack of ground truth.

**Bias in $A$.** We next consider issues regarding the protected attribute ($A$), e.g. the group-level attributes used for disparity assessment, such as race, sex, and other designations. Even when the protected attribute information appears, it is often estimated under noisy measurement processes. This can be seen in Lum et al. [38], one study which provides descriptive statistics on how officer-reported racial designations in CJ data were inconsistent for all but those people designated as Black.

---

[3]Stack v. Boyle, 342 U.S. 1 (1951), United States v. Salerno, 481 U.S. 739 (1987)

Hanna et al. [39] further discusses challenges and inconsistencies associated with demographic data. Notably, COMPAS race categories lack Native Hawaiian or Other Pacific Islander, and redefine Hispanic as race instead of ethnicity. Racial categories largely go unquestioned despite instability well-documented in intersectional work [40, 39]. All this obscures an understanding of whether and how an RAI functions differently by race.

Some methodological approaches suggest using proxy measures (e.g. covariate-based proxies such as the distribution of race by geographic area, or viewing observed $A$ as a noisy misclassification of the true value) to estimate disparities when individual level-observations of the protected attribute are unavailable or noisy. Unbiased recovery with covariate-based proxies is impossible [41]. Wang et al. [42] studies robust evaluation and classification. Lahoti et al. [43] consider an adversarially reweighted approach. Lamy et al. [44] show that under a specific noise model, original fairness constraints can be guaranteed for the mean-difference score criterion by increasing a constraint violation parameter. Awasthi et al. [45] provide conditions on whether the equalized odds post-processing constraint imposed on noisy $A$ still at least improves equalized odds (TPR and FPR). In the CJ domain, accurate point estimates of disparities are the ultimate target. While directional analysis of disparity improvement may be more useful in other settings, such analysis may be less helpful for CJ. Domain-specific empirical studies of the extent of misreported $A$ in CJ may be needed.

**Bias in $X$.** Many of the covariates used in pretrial RAIs are low dimensional summaries of criminal history, such as past arrests or convictions. Since these measures inherit the racial bias of local law enforcement, they may perpetuate these same harms [46]. To quote Sandra Mayson on her experience as a public defender in New Orleans, "If a black man had three arrests in his past, it suggested only that he had been living in New Orleans. Black men were arrested all the time for trivial things. If a white man, however, had three past arrests, it suggested that he was really bad news!" [17].

Additionally, there are many individual points of discretion for data entry including assessor discretion with evaluating individuals, booking officers and prosecutors, to final bail decisions made by judges. For example, COMPAS asks "Is there much crime in your neighborhood," [47] which allows racialized meanings of crime [46, 48, 49] to influence the interpretation. Some RAIs incorporate unvetted booking charges that are subject to officer discretion [38]. Khani and Liang [50] show noise in covariates leads to statistical inconsistency and group-specific bias, even for ordinary least squares.

Finally, data processing that is done before a CS researcher ever touches a dataset hides ethically complicated decisions. For example, should past convictions for crimes that are now decriminalized be counted in the number of past convictions? Should there be a "sunset window" on past FTAs, after which they can't be considered by a RAI? All of these choices have statistical implications: Friedler et al. [51] show that encoding choices have implications for the resulting fairness reported. Johnson et al. and Yukhnenko et al. [52, 53] study proximal/recent/dynamic risk factors vs. static/historical risk factors (criminal history), the former are as predictive as the latter. Given the idea of *predictive multiplicity*– that multiple different predictive models achieve similar aggregate accuracy [54] – different data processing decisions may better realize normative objectives like not penalizing now-decriminalized behavior or not allowing one's past mistakes to haunt them forever. Many ML models admit the same accuracy and the specific ones used may have been tie-broken on these previous considerations. While current datasets focus on disparities measured at a point in time, longitudinal data may be required to investigate the impacts of compounded qualitatively salient inequities.

**Issues with the distribution of data.** CJ data is collected downstream of previous decisions: indirect structural forces, such as criminogenic contacts with the CJ system (e.g. increasing number of prior charges), or direct endogeneity, via differential location-based policing and/or arrest practices that induce differential screening intensity. This introduces selection bias [55, 56]. Beneficial CJ reform may induce systematic shifts in distribution for future populations subject to RAI prediction. While domain adaptation and distribution shift are well-studied in statistics/CS/ML, true distributions are ill-defined in CJ. RAI datasets suffer from *selective labels*, e.g. that outcomes ($Y$) recorded in datasets are only recorded post-selection into the dataset in the first place since observing re-arrest or non-re-arrest is only possible for individuals not detained [57]. Racial disparities in detention or bail decisions induce systematic mismatches in the underlying distribution relative to the universe of those who had interacted with the CJ system. Kallus et al. and Rambachan et al. [58, 59] study conditions when these distributional differences lead to biases in incomplete fairness adjustment, although normative implications depend on the joint utility associated with censorship in/out of the dataset and the final positive outcome of interest. Singh et al. [60] study conditions (assuming a given

Table 1: Issues with the COMPAS Dataset

| Issues with COMPAS Dataset | CS translation |
|---|---|
| Arbitrary choice of threshold for assessing classification disparities | Incorporate discretion |
| 2-year follow-up period includes arrests after case closed (no longer pretrial) | Bias in outcome value Y |
| Differing follow-up periods for rearrested vs. not rearrested individuals | Bias in outcome value Y |
| Restricted range of risk scores | Issues with distribution of data D |

causal graph) under which a stable predictor, in the sense of accuracy and fairness, could be learned. Coston et al. [61] develop counterfactual risk assessments to account for decisions. Further empirical and methodological work on these distribution shifts, and decisions that shift distributions, would be valuable; as well as translational work to further communicate these issues to CJ practitioners.

Table 1 uses the above framing to taxonomize a few COMPAS data issues, with details in A.2.

## 4 Limits of Algorithmic Fairness in RAI Outputs for Real-World Outcomes

Algorithmic RAIs are part of a complex socio-technical system of decision-makers and institutions that introduce multiple *points of discretion* into what algorithm designers often think of as an automated, single-decision system. As a result, focusing only on algorithmic fairness is not enough to ensure that fair decisions are made based on the predictions of risk produced by algorithms. Even if the data bias issues discussed in Section 3 were not present, claims of real-world fairness derived solely from algorithmic fairness cannot be made because the application of the RAI also impacts outcomes. A range of legal actors use the predictions produced by RAIs to make decisions so their interpretation of those predictions, and their decisions about whether to use the predictions or not, play a large role in getting from fair algorithm to real-world fairness. We provide two examples below.

One major downstream point of discretion is judicial discretion, where judges choose how to interpret pretrial RAIs and issue decisions. The degree to which judges follow RAI recommendations varies greatly, with most jurisdictions giving judges final discretion but some having automatic decisions for certain risk scores (e.g. automatic pretrial release for "low-risk" defendants in Kentucky) [12]. RAI critics point out that RAIs, even if developed with the intention of "fairness," can be ignored by unfair judges, and RAI proponents point out that the improvements in outcomes that could be achieved with RAIs may be lost when judges override or ignore RAI recommendations. Studies have shown that even with an explicitly stated goal to reduce jail and prison populations, many judges ignore RAI recommendations [62]. Studies have also suggested that there is a racial component to deviation and that judges may be more likely to deviate from predictions for Black than white people [63, 64] and for "detain" recommendations than "release" recommendations [65], but it is not yet clear how such deviations relate to more or less disparities in detention outcomes. Since imposing oversight on judicial decision-making is notoriously difficult, judges have nearly unchecked discretion [66] and have resisted measures such as mandatory written responses about deviating from RAIs [67].

Discretion also extends beyond judicial decisions. Jurisdictions have discretion in mapping numeric RAI scores and probabilities to recommendations for decisions. Although the Public Safety Assessment mandates reporting risk via the scaled scores, jurisdictions determine thresholds for recommendations of varying degrees of unsupervised or supervised/conditional release in the release conditions matrix. This allows jurisdictions to account for local context but also introduces discretion and wide variation in the *absolute risk* associated with release conditions, with one study reporting predicted probability of re-arrest ranging from 10% to 42% for "high-risk" individuals across RAIs [29]. Jurisdictions have also implemented RAIs that contradict implementation guidelines.[4]

---

[4] For example, the Public Safety Assessment guidelines suggest that it should never be used to recommend detention (Release Conditions Matrix 2020), yet when Cook County, Illinois adopted it in 2015, those with the highest risk score were recommended for pretrial detention, continuing the high detention rates for "high-risk" defendants [68].

These points of discretion illustrate the role that human judgement plays in translating the RAI prediction into a real world decision. It is critical to recognize that the application of a RAI has as much of an impact on fairness as the algorithm itself, if not more, because the fairest algorithm will not result in fair decisions if it is not used or it is misused by the ultimate decision-makers.

## 5   Normative Values Embedded in Use of RAI Datasets

When researchers engage in work with RAIs used in the criminal legal system, at pretrial or other time points, they are not just contributing to a discussion about how to develop a fair algorithm, but also a much larger discussion on the role of values such as fairness, justice, equality, and power in the criminal legal system. Researchers in the space must grappled with the broad ethical implications, ranging from implications on the quantitative notions of fairness to the structural conditions that perpetuate inequality. We highlight some of these implications in the following assumptions and decisions made in benchmarking: 1) CJ tasks and outcomes, 2) fairness definitions, and 3) real-world CJ reform and societal impact.

**Tasks and Outcomes.** When inheriting tasks and outcomes from previous experiments to benchmark "performance," implicit normative considerations may slip through the cracks. Assumptions and encoded values of CJ predictions start but do not end with the data. Decision-making, actuarial or clinical, includes criminal history as an input: not only is this subject to technical concerns as discussed in Section 3, it places value on the fairness and justness of individual decisions, laws, norms and institutions. Upholding current norms and legal definitions around crime risk in choosing outcomes is also value-laden. By optimizing for an outcome of reduced crime risk, algorithms currently primarily favor incapacitation of individuals to prevent and deter crime, instead of retribution or rehabilitation [6]. Primarily valuing reduced crime risk may also come at the cost of overvaluing incarceration, as incarceration by definition reduces one's likelihood of committing a crime in the near future [69], and create substantial lasting harms to those incarcerated and their communities [70, 71]. Even short periods of incarceration have such severe consequences that some studies suggest a person must pose an extremely high risk of serious crime in order for detention to be justified [28, 72].

**Statistical Fairness Objectives.** As discussed in Section 4, statistical fairness objectives are limited in their ability to directly translate to outcomes that align with their predictions due to real-world decision-makers. There also exists a disconnect between statistical fairness objectives and fairness as a social ideal, partially due to the values of fairness objectives and partially inherent to the contexts in which RAIs are deployed. The first component of the disconnect is a result of fairness objectives each uniquely encoding political and ethical values, when specific measures often cannot simultaneously be satisfied [73] due to unequal circumstances [74, 69]. Relevant ethical considerations, such as whether outcomes or treatment should be equal, what equality consists of, and axes of fairness/sensitive attributes, are also relevant for human-based decision-making, though perhaps more implicit in prediction-based decision-making under a guise of objectivity. Secondly, fairness objectives are inherently limited in the degree to which they translate to fairness as a social ideal, which requires placing the algorithm in context of the unjust institutions they are a part of. Fairness as a social ideal is unstable and uncertain, constantly being debated and shaped, "a process of continual social negotiation and adjudication between competing needs and visions of the good" [75].

**Real-World Impact.** Tying CS research that uses RAI data to real-world impact implicitly assumes that RAI reform is an avenue for CJ reform. Determining specific avenues of change around theories of decarceration, decriminalization, bail reform, abolition, etc. casts value judgments on what a more ideal and just world looks like, which is inseparably tied in an understanding of how historical structures have reinforced injustice. Even the notion of structural reform, for policing, incarceration, or CJ in general, is itself value-laden and a contested theory of change [48, 76]. Contributing to CJ reform values harm reduction in the present at the cost of legitimizing existing structures as worth reforming and/or possible to reform. Our point is not that one position is more appropriate than another, but rather that engaging with CJ reform via the development of algorithmic RAIs *necessarily* places the algorithm designer in the middle of a debate over values that they cannot avoid.

## 6   Established Norms in Fields Engaged in Criminal Justice Work

RAI datasets have been used for decades in fields that have historically engaged in CJ work (e.g., criminology and psychology). Knowledge of the existing standards and methodological practices in these fields could provide AI/ML researchers with important insights into publicly available CJ

datasets. In this section, we discuss the process of creating datasets, standards for RAI publications in other fields, guidelines for RAI use in practice, and updated standards for language used in CJ publications. We also provide specific takeaways within each of these topics for AI/ML researchers.

**Data Collection.** A great deal of time, effort, and expense goes into the creation of the CJ databases that are available online [77]. It may take years to develop relationships with relevant stakeholders to obtain data access. This process is complicated by political or pragmatic concerns and motivations around data access. Stakeholders often require a data use agreement which may restrict the data researchers can use and whether they can share it beyond the scope of the current project. The administrative records that contain CJ-related data (covariates and outcomes) are stored across multiple systems and agencies with varying identifiers, granularity, and accuracy, requiring intricate record linkage [78]. Management systems are often outdated and commonly require information to be manually entered by staff, introducing opportunities for errors. Finally, some information is unavailable in administrative data but must be collected via interviews with people who are justice-involved [79, 80]. Interviews add further complexity, resource, and potential sources of biases, including self-selection, non-response, impression management, rater bias, and attrition, among others [81, 82, 83]. Researchers must go through a rigorous process to obtain approval to interview people who are justice-involved [84] and they may be limited in the length and content of interviews.

To avoid making assumptions about the data, AI/ML and other technical researchers should contact CJ researchers, particularly those responsible for datasets, to discuss decisions made during data collection and dataset creation. Additionally, due to the availability bias in what data is publicly accessible, generalizations drawn from inferences on these datasets must be viewed with caution. Finally, we encourage AI/ML researchers to consider collaborating with CJ researchers to obtain data and create datasets because CJ researchers offer important contextual and practical knowledge while AI/ML researchers have data organization/management skills CJ researchers likely do not, especially when dealing with large datasets.

**Standards.** We turn next to the issue of standards in publication. RAIs have been used in CJ and health care contexts for decades by psychologists, psychiatrists, and others [85, 86, 87]. When psychology researchers examine RAIs for accuracy or bias, they typically follow evaluation standards created jointly by educational and psychological research associations in compliance with APA ethical guidelines, which serve as the consensus empirical criteria of test bias assessment. Many RAI bias studies apply these standards [88, 89, 90, 91, 13] to consider instrument performance across the full range of scores rather than binary or categorical classifications. Psychologists are also informed by methodological primers [92, 93], which recommend statistical tests to assess discrimination (i.e., how well an instrument separates people who do and do not experience an outcome of interest) and calibration (i.e., how well predicted risk agrees with observed risk). Notably, psychology uses a different definition of calibration than ML (same likelihood of an outcome across different groups of people) [94, 95]. One final source of methodological guidance is the Risk Assessment Guidelines for Evaluating Efficacy Statement [96], a list of 50 items that should be included in publications on studies of RAIs, intended to improve comparability across studies and transparency in reporting on research findings. Notably, all these standards and methodologies are most widely used in the field of psychology, with other fields (e.g., criminology) adhering to them to a lesser degree [13].

These standards are reflected in decades of research on RAIs. AI/ML researchers should be cognizant of these standards because they will clarify some of the decisions that are being made in research coming out of fields like psychology and criminology. Further, the Risk Assessment Guidelines for Evaluating Efficacy Statement functions similarly to a sheet for a dataset [19]. Finding publications that adhere to this standard would provide AI/ML researchers with important information about the underlying data.

**Guidelines.** There are established guidelines regarding how RAIs can and should be implemented to inform decision-making: selecting an appropriate RAI, implementing the instrument with fidelity, and making decisions informed by RAI results. [97, 85]. They can be instrument-specific, discuss fairness, transparency, and effectiveness, [98, 99], and make dataset determinations. For example, pretrial RAI research studies use a one or two year cutoff for the follow up period because pretrial RAIs are only intended to be valid during the pretrial period of at most two years [100, 101]. Importantly, these guidelines are often rooted in ethical or legal obligations [102, 103]. Familiarity with guidelines is important for AI/ML researchers because using data from RAIs without understanding how they are actually used in practice could lead to false conclusions. Further, new research that contradicts the

way RAIs are actually being used is unlikely to see uptake in practice. Finally, recommendations based on new research may be impossible to apply if they contradict current guidance rooted in ethical or legal obligations.

**Language Guidelines.** AI/ML research papers should consider updated language guidelines when writing about the CJ system. Several guides outline outdated vs. currently preferred terms: the intent is to stop using terms that define people by their crimes and punishments and instead center their humanity (Cerda-Jara et al., 2019; Solomon, 2021). Some of these guides are specific to people who are justice-involved [104] while others cover biased language more broadly (e.g., APA Standards for Bias-Free Language). The preferred terms for talking about people who are justice-involved are shifting away from previously acceptable but dehumanizing terms. Where before papers used terms like "felons" and "prisoners," now phrases like "person convicted of a felony" or "incarcerated person" should be used. In the quest for fairness, researchers can unintentionally perpetuate unfairness if they use language that contributes to the marginalization of the very people for whom they are trying to create more fair algorithms.

## 7 Mismatch Between AI Fairness Practices and Criminal Justice Research

Following our discussion of the many challenges and pitfalls of working with RAI datasets, we analyze how current research and publication practices in algorithmic fairness can be ill-suited for meaningful engagement with fairness in CJ applications and can exacerbate previously delineated issues with data quality, real-world relevance, and inadvertent normative implications. We highlight a focus on methods, the decontextualization of data in experiments, and the conference publication workflow as factors contributing to this mismatch. In addition, we note how problematic experiments can become adopted as benchmarks, creating a compounding effect. Our critique focuses on *methods papers* (justifying a particular algorithmic measure or method on an RAI dataset) instead of "science" or substantive papers that focus on learning new science from data with typically standard methods (e.g. studying issues within specific RAI datasets).

Firstly, we argue that the focus on methods in the ML research literature, while well-suited for the ML community, is misaligned with the broader scientific goals involved with studying RAIs. A methods paper typically justifies a method in relation to other methods, rather than justifying it in terms of new science output. The dataset is secondary – merely a benchmark to provide a baseline comparison. The methods paper must also then model the science goal – say of greater equity in the RAI – as a measurable objective that it can outperform the competition in. This leads to an often obsessive focus on SOTA (state-of-the-art) optimizations in which the data remains passive. A paper can be of high quality in a pure AI/ML methods sense, but irrelevant for CJ impact or worse, introduce or perpetuate mistranslations of the CJ context. Given the visibility and stature of ML venues, this can have inadvertent consequences as methods claims are vetted far more than CJ-adjacent claims.

Secondly, placing the data in subservience to optimization goals decontextualizes it – the objective is beating a measure of performance instead of gleaning new insights from the data. This leads to many of the problems described in previous sections, in which meaningless or erroneous conclusions are drawn from the data due to a lack of data context. For example, experiment designs are sometimes so disconnected from context that a prediction of high-risk is treated as the preferred outcome for an individual in an equal opportunity model simply because high-risk is the "positive" label in the dataset.[5] It is also not uncommon for "granted bail" to be described as the positive decision [105] despite the fact that many individuals are unable to afford bail [106, 107, 108, 109], implying that being incarcerated during the pretrial period is positive for them. Decontextualization of the data creates further problems when algorithmic fairness papers imply that their results have consequences for how RAIs work (or should work). Risk assessment in CJ is not a modular pipeline in which each component can be replaced with a fairer version the way you would replace a sorting algorithm with a more efficient implementation. It is a tangled mess drenched in an ongoing history of inequity. Claiming improvements based on using an RAI dataset out of context can result in a host of issues that challenge the validity of conclusions drawn and raise ethical questions about claims made.

Thirdly, the conference publication workflow and community norms in CS make appropriate use of RAI datasets difficult, in contrast to other fields that rely on journal publications. Experiments in AI/ML conference papers are often implemented quickly and for the purpose of supplementing

---

[5] An author of the present work has noticed this several times in reviewing manuscripts for top-tier ML conferences.

the methodological or theoretical contributions of a paper that is already intended for submission or even under review. For example, researchers sometimes perform new experiments on a new dataset during a rebuttal period of less than one week, which favors grabbing any freely-available benchmark datasets containing people as data points and demographic labels for those people. For instance, the COMPAS dataset has emerged as a popular choice satisfying these criteria and bolstered by its connection to the highly-cited *Machine Bias* article [1] and status as a "real-world" dataset.

Finally, the importance of benchmarking from a methods-first point of view causes inappropriate experiments to have a broader, compounding effect on future research. Use of a dataset for empirical validation in a seminal paper often leads to follow-up work replicating those experiments and establishing that dataset as a de facto benchmark for the problem studied. Expectations of the typical conference review process, which limit discussion and author rebuttal relative to journal publication norms of other fields, reinforce this positive feedback loop. Omitting a commonly-used dataset in an experiments section is a dangerous gamble for authors whose paper may be rejected for failing to compare to established benchmarks. Thus, researchers are incentivized to continuously replicate even flawed experiments. This is not a purely hypothetical concern: this compounding effect is one of the reasons for the (in our opinion) overuse of the COMPAS dataset.

Calls to address these issues often implore researchers to collaborate with domain experts and practitioners, but these collaborations are challenging. The rapid publication cycle in which methods might build upon each other within a year or two often establishes an appearance of consensus around a benchmark, which becomes difficult to change or refute later. This cycle does not match the slower, more deliberate process of acquiring and understanding data from a single source. Furthermore, the fundamental difference between science and methods contributions needs to be recognized as well as the focus on data-as-benchmark versus deriving meaning from the data.

While none of these issues are insurmountable, researchers who wish to engage meaningfully in this space must ensure that their incentives are aligned in order to avoid the problems we enumerate.

## 8 Call To Arms

We now suggest best practices for addressing the challenges of relying on RAI datasets for broad investigations of fairness as well as CJ-focused algorithmic interventions.

**Things not to do.** Firstly, researchers working broadly in algorithmic fairness should avoid the use of CJ-related datasets (e.g., COMPAS) as generic real-world examples to illustrate or benchmark a new fairness algorithm or measure. As this deep dive into CJ datasets has hopefully shown, CJ data should be interpreted within a rich domain context, and taking it out of this context to perform a typical 'horse-race' analysis is misleading. Even worse, such analysis could be misunderstood as saying something specific about CJ problems, which would be incorrect as well as politically fraught.

Researchers should also avoid making broad conclusions about practices in CJ solely from the use of CJ datasets. Once again, the context in which these datasets can be interpreted varies widely even within different locales with different laws, practices and data acquisition methods. Broad conclusions are likely to be misleading or wrong, and risk becoming political scoring points against the broader backdrop of discussions about CJ and ideas for reform.

**Things to be careful about.** This paper does not advocate completely avoiding datasets from CJ or quantitative methods for real-world problems. Rather, we encourage understanding of the CJ context and how such models will be used in practice to foster meaningful progress. By partnering with CJ researchers, ML researchers could gain access to insights about the data and challenges to model use in practice that may inspire new research directions and prevent falling into the traps previously identified. ML researchers working in this space should particularly avoid over-selling the implications of or potential uses of their work. Thorough discussions of modeling assumptions are useful to communicate the work to people from different disciplinary backgrounds. Clearly and thoroughly describing limitations of the methodology as well as potential consequences of deploying the system in the real world will go a long way towards a more meaningful engagement.

Some of this may be hard to do within a traditional CS conference publication workflow and timetable, where the applying a method to a new dataset at the last minute is acceptable. Accessing CJ datasets often requires applying months in advance for approval, followed by data cleaning and interpretation, and would contribute to studying disparities in other RAIs besides COMPAS. Along similar lines, conference organizers and reviewers can enforce requirements for ethical/acceptable

use, documentation, and release of CJ datasets in particular. If researchers use COMPAS out of a perceived expectation to use a real-world dataset, even if benchmarking on that dataset is irrelevant as we argue in Sections 3 and 4, then reviewers should encourage well-designed simulations instead.

**Things that might be helpful.** Even when meaningful collaborations with domain experts in CJ are not possible, ML researchers can leverage their specific skill set and expertise for the broader issue of RAIs and their use. We list a few greatly-needed examples of such contributions.

*Building datasheets and model cards for CJ datasets/RAIs.* Datasheets [19, 110] are becoming a valuable accountability mechanism to understand the limitations and operating conditions for any dataset. Preparing datasheets for CJ datasets will require capturing the choices described in sections above as well as many of the lenses on bias that ML researchers have developed over the years.

*Identifying implicit assumptions in fairness algorithms.* A critical (and reflexive) examination of fairness interventions is a process by which the researcher identifies value positions and assumptions inherent in the way a particular intervention is formalized, or how a measure of fairness is formulated (E.g., whether the algorithm falls into a particular kind of formalization trap or whether its framing of fairness is based on assumptions about the underlying data). Given the many contested assumptions and values that stakeholders bring into the discussion of any RAI, surfacing such biases ensures that algorithms are not used blindly in a way that could exacerbate existing biases.

*Investigate where problems in data and interpretation could make algorithms fail.* Even if we can surface the underlying assumptions that an algorithm uses and verify that these are reasonable within the context, can we be sure that the algorithm will behave in a robust manner if the assumptions are violated slightly? This is the idea of assumption stability – it asks if algorithms are robust under changes in the underlying assumptions (which might manifest as distribution shifts or unexamined correlation between features). The ML and algorithms communities have a rich history of robustness investigations in the context of learning models, and their tools can aid in addressing this question.

*Investigate the use of different metrics and their problems.* There are many standard metrics used in the risk assessment literature to validate RAIs. For example, a model that gets an AUC score of above 0.65 is considered a reasonable model for deployment [79, 111]. However, as Zhou et al. [112] have shown, reporting accuracy disparities as group-conditioned AUCs does not actually measure across-group ranking disparities and different models with the same AUC might perform very differently across demographics or risk thresholds. Similarly, work by Marx et al. [54] has identified ways in which models that exhibit the same overall accuracy might behave differently on individuals, and proposed a new measure of predictive multiplicity to capture this variation as a way of introducing reasoned skepticism about the performance of a given model. Similarly, testing standards in other fields require examining the functional form of the statistical association between RAI scores and predicted outcomes [113].

*Expand limitations of benchmarking for real-world relevance.* Our concerns around RAI datasets center around relevance: representativeness of the real-world and applicability to impact real outcomes. Our work enables future work to analyze how RAI datasets are ill-suited for benchmarking in other ways, shape criteria for evaluating benchmark datasets, and explore if benchmarks for socio-technical systems like the CJ system can exist at all in a beneficial and ethical way.

## 9   Conclusion

Throughout this paper, we have argued that using the COMPAS dataset as a generic benchmark, perhaps in deference to convention, is a poor choice for technical and normative reasons. Conducting CJ research necessitates developing a contextual understanding of the past and present of the CJ system, which demands grappling with questions around ethics given deeply entrenched systemic inequity. We are encouraged by the increasing awareness within the CS discipline around broader impact, negative consequences, and ethical considerations of this work, as well as a growing understanding of the elaborate ways that algorithms interact within socio-technical systems. We hope that this paper can provide insight into issues and complexities with using RAI datasets as benchmarks for real-world fairness, to better align with goals of developing more informed and just policy decisions.

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

**Acknowledgments**  Angela Zhou acknowledges support under NSF 1939704. Samantha Zottola acknowledges support under the John D. and Catherine T. MacArthur Foundation's Pre-Trial Risk Management Project.

## Checklist

