# OpenReview forum: "It's COMPASlicated: The Messy Relationship between RAI Datasets and Algorithmic Fairness Benchmarks"
_NeurIPS.cc/2021/Track/Datasets_and_Benchmarks/Round1 — NeurIPS 2021 Datasets and Benchmarks Track (Round 1)_

### Official Review · Reviewer_4EMA · 2021-06-24
**Review for paper**

**Rating:** 8
**Confidence:** 4
**Clarity:** Yes

**Strengths:**

- Very strong focus on the foundations/sources of CJ datasets
- Illuminating takes on the differences in metrics in CS and other domains
- Good points about the conference cycle and the impetus towards doing benchmark studies

**Weaknesses:**

- As discussed above, would really love to see a case study, specifically show the pain points in the COMPAS data as a summary
- Citations could be improved here and there, I provided some suggestions

**Additional Feedback:**

-

**Correctness:**

Submission is neither a dataset nor a benchmark. Other claims made in the paper hold up.

**Documentation:**

N/A

**Ethics:**

The ethics discussion in the paper is satisfactory.

**Relation To Prior Work:**

Yes

**Summary And Contributions:**

This paper highlights the problems of overused datasets in the field of fair machine learning, especially when such datasets pertain to the field of criminal justice (CJ). I commend the authors for choosing this problem to expand on, as the COMPAS dataset released by ProPublica has become the de-facto benchmark for almost every paper describing a technique on fair classification. I particularly enjoyed the discussions about the different metrics used by Psychology than CS for CJ datasets, the specific focus on the pitfalls of method papers wrt conference cycles and blindly trying to apply those ideas to sensitive CJ datasets.

My main point of contention is that in my opinion, this paper is lacking an empirical analysis or even a value-based, methodological analysis of the pitfalls of the COMPAS dataset (since it is the most representative of the widely used CJ datasets). The abstract made it seem like this would exist somewhere in the paper, and yet after going through the entire paper (and the appendix), it seems like an actual dataset discussion, for instance, a table summarizing the data distribution, the features -- and what how these features were defined by CJ folks vs how they are misinterpreted by computer scientists without domain knowledge, is missing. I am aware that this has been critiqued widely in the past, and that the authors have profusely cited a lot of previous literature which looks into this, but considering the goals of the Datasets and Benchmarks Track, this paper narrowly fits into the "Frameworks for responsible dataset development" silo. As described above, a table or section- even somewhere in the appendix - would make the paper several times stronger because it would add an audit and fit the venue much better. Fortunately, this is addressable and I am willing to give the paper the high score it deserves if the authors do so.

Other nits:

- "For example, it is not uncommon for the experiment design to be so disconnected from context that a prediction of high-risk is treated as the preferred outcome for an individual in an equal opportunity model simply because high-risk is the “positive” label in the dataset." Citations?
-" It is also not uncommon for “granted bail” to be described as the positive decision, implying that the many individuals who are unable to afford bail [citations] being incarcerated during the pretrial period is positive.". I had to read this sentence several times and then search about it to understand the (brilliant) point that the authors have made. Point being, this sentence assumes a lot of domain knowledge about the US prison bail system from the reader and could be reworded or expanded to make it slightly more accessible.
- Section 6 and Section 8 are both guidelines, and I wonder if they could be reworded/rearranged to highlight their individual contributions more. Perhaps rewording section 6 as established norms and section 8 as proposed guidelines (although call to arms is great too) might be better?
- Citations assume that CJ is almost entirely used in classification problems. Unfortunately in my experience, they are also retrofitted by authors in other use cases like ranking, clustering, recommendations, etc. Citations on those and talking briefly about those harms might be great too if it can fit the narrative of the paper.
- "Some methodological approaches suggest using proxy measures (e.g. covariate-based proxies such as the distribution of race by geographic area, or viewing observed...". There's a bunch of empirical and policy work you could cite here, for instance
      - https://arxiv.org/pdf/1912.06171.pdf
      - https://arxiv.org/abs/2011.02282
      - https://arxiv.org/abs/2105.02091
      - https://www.latimes.com/business/la-fi-rand-elliott-20160824-snap-story.html

---

### Official Review · Reviewer_aHiH · 2021-07-03
**Raises very important questions on benchmarking in fair ML**

**Rating:** 9
**Confidence:** 3
**Clarity:** The paper is a pleasure to read as it…

**Strengths:**

The paper provides a comprehensive analysis of the potential issues that arise when using RAI datasets to e.g. test new methods. It brings together work from CS, CJ, psychology and other fields to provide this overview. This intersectionality is extremely valuable as it offers a perspective on how CS’s usage of RAI datasets differs from that of other fields - which have been using RAI datasets for much longer than CS. These insights are particularly helpful for researchers who are considering using e.g. the COMPAS dataset, but haven’t thought through what the consequences of this might be. It can thus create an increased awareness of potential ethical issues. Lastly and most importantly, the paper raises very interesting questions and could start a much-needed debate about the usage of real-world benchmarking data that comes from socio-technical systems.

**Weaknesses:**

The paper covers a lot of topics and for the most part achieves to do this in an accessible manner. However, some parts appear too condensed (potentially due to the space limit) and could profit from a bit of elaboration, in particular:
- lines 103-105: The explanation of construct (in)validity in recidivism predictions could profit from a brief definition of construct validity as CS researchers are in my experience not familiar with the concept unless they are familiar with [27] and related work.
- Methodological approaches regarding the reduction of biases could be explained in more detail. For example, “Wang et al. [41] studies robust evaluation and classification.“ (line 140 f.) might be a helpful reminder to readers familiar with that work, but otherwise appears difficult to follow.


**Additional Feedback:**

Here is a list of typos / smaller issues I noticed while reading the paper:
- reference [122] could be updated to the FAccT paper, which should include Maria De-Arteaga as an author
- line 81: is there a word missing? “ML studies have mainly explored the former through “benchmark misalignment” which arises with systematically unstable labels [22, 23], but misalignment in the context of socio-technical systems is *rarely(?)* discussed.“
- reference [27] could be updated to the FAccT version
- delete one “in the first place” in line 177: “RAI datasets suffer from selective labels, e.g. that outcomes (Y ) recorded in datasets are only recorded post-selection into the dataset in the first place since observing re-arrest or non-re-arrest is only possible for individuals not detained in the first place [55].”
- line 187: institution → institutions
- line 237: still stands and is → still stand and are
- line 243: tends to favor → tend to favor
- line 243 f.: favor equality of process of deciding an outcome rather than the outcome itself → favor equality in the process of deciding an outcome rather than in the outcome itself

Personally, I’d be very curious about the authors’ thoughts on how simulated data could be helpful in addressing the discussed issues, but I recognize that this might be difficult to add due to page limit constraints.

**Correctness:**

The claims made in the paper are backed up through research from various fields, such as CJ and psychology.

**Documentation:**

Not applicable as the paper neither presents a dataset nor a benchmark, but criticizes the way in which the community is currently engaging with RAI datasets.

**Ethics:**

The paper highlights highly relevant ethical questions and attempts to create more awareness among CS researchers for the potential negative societal impacts of using RAI datasets in an unreflected manner. I do not see potential for the paper to cause negative societal impacts, but rather to mitigate it.

**Relation To Prior Work:**

To my knowledge, previous contributions have rarely looked at the potential issues of using RAI datasets in experimental CS studies. The paper draws on existing work to demonstrate these issues. This is done in a clear manner.

**Summary And Contributions:**

This paper discusses the usage of RAI datasets (such as the COMPAS datasets) in CS research. It highlights what issues may arise when using such datasets and critiques their usage for benchmarking purposes, which typically comes at the cost of understanding the data’s context. This leads both the reader and the paper to ask whether RAI datasets can even be used in an ethical manner in CS research.

The paper covers a lot of ground and discusses the following topics:
- biases in the dataset (X, Y and A)
- differences between deploying RAI systems in theory and in practice
- implicit consequences of using RAI datasets (e.g., legitimizing current CJ system instead of working towards reform)
- how other fields work with RAI datasets and what CS researchers can learn from them
- critique of research practices in ML fairness (e.g., focus on methods papers and the disconnectedness from the data context)
- suggestions for what to do about these issues in the future

---

### Official Review · Reviewer_qVFn · 2021-07-05
**Integrated and interdisciplinary perspectives are needed to understand how sensitive datasets are used as benchmarks in fair ML**

**Rating:** 8
**Confidence:** 5
**Correctness:** Solid reasoning and arguments, deep d…

**Strengths:**

The most significant contribution of this work highlights the importance of the criminal justice context, the popularity of ("fair") ML, and the ubiquity of the COMPAS dataset---and how failing to consider the context and values of RAI datasets creates new risks and harms. This is even more true as "fair" systems can exacerbate biases and harms.

This work shows the context and values embedded in using RAI datasets for benchmarking. This paper draws from lessons from criminal justice and psychology and shows how these are present in the fair ML use case, even if they are not explicitly considered. This work shows the structure of benchmark problems in fair ML and where understanding from criminal justice, psychology, and science and technology studies can help us understand where values and harms are embedded. This benchmark problem structure is combined with some technical references (although see notes about S3 in weakness).

If this paper is (lightly) improved to better integrate its technical discussions, this paper could provide an important warning and call to arms for fair ML, as well as lay out an interdisciplinary strategy to explore benchmark datasets from other domains in fair ML.

Some additional key strengths:
- Reflects deep disciplinary engagement with criminal justice research.
- Provides a framework for understanding the structure of benchmark problems, and where it may not align with practical use.
- Provides important warnings about such use, particularly given the ubiquity/importance of RAI dataset use and the lack of engagement with the values implicit in such use.
- Provides guidelines for the fair ML community (dos and don'ts), some weaknesses of the fair ML community (S6), and some other suggestions for practical use (S5).

**Weaknesses:**

Lack of integration or synthesis: Several areas of the paper list technical ideas/results without connecting them to the rest of the paper. This paper does important bridging work, so providing this type of insight would be a major contribution; without it, this paper won't achieve as much in this venue or from this audience.

- In section 3, every subsection ends with a list of technical results without interpretation. Are we supposed to understand that these results solve the problems just introduced? Or are they supposed to show that the field is/is not answering the appropriate question? What should be happening instead, or what work is needed to make this more/less useful? Given the bridging work this paper is supposed to be doing, providing this type of insight would be a major contribution; without it, this paper won't achieve as much in this venue.
- Section 4: This section is about limitations of "fairness," but seems quite focused on points of discretion. How should the reader understand how this section fits in with the rest of the work? Do the authors intend to make a measurement argument (in which case it belongs better integrated with S3) or is the primary argument about limitations of mathematical/algorithmic definitions of "fairness"? Later references to this section as the latter, but the section does not seem focused as such.
- Section 5 contains some of the core arguments about values that are relevant for this audience; can any of these ideas be integrated earlier or combined with section 4? In addition the section on "fairness objectives" seems very important for this paper, but it is short and the last two sentences are not clearly written.
- Section 6 has good content but it is hard to see the takeaways, and it feels out of place. Can these ideas be combined elsewhere, moved into background, or even just swapped with section 7?

Some ideas are introduced without context, and new terms are sometimes introduced unnecessarily instead of stating the intended conclusion. This will hinder uptake of these ideas by the fair ML community. Some salient examples:

- S3: what is "measurement bias" vs. measurement error vs. other types of threats to validity? Are these problems with the construct itself or its operationalization [re-arrest as re-offense]?
- S3: Does the term "label bias" mean "bias in the assignment of labels"? If so, then here and in similar instances, new terms appear to be introduced but not defined, and instead could be said more clearly.
- S3: Predictive multiplicity makes sense in its later context, but it is less clear what it is adding here
- S3: "may contribute to distributional harms" -- this could have multiple meanings, and the use of "distributional" is especially unclear for ML audiences -- please translate this point for a broader audience
- S2: relevance, solvability, bioinformatics -- this last sentence is not clear - what is this accomplishing for us?

For S3, two recommended citations for demonstrating different types of lack of validity/biases, particularly for Y and X:
- A Roberts. 2018. Arrests as guilt. Ala. L. Rev. 70
- AL Hoffmann. 2019. Where fairness fails: data, algorithms, and the limits of antidiscrimination discourse. Information, Communication & Society

The conclusion seems to be written for researchers who identify as doing criminal justice research, but the target of the paper seems to be anyone using RAIs. Your claim might be that these are actually not so distinct, but this conclusion doesn't make that claim. Some of the stronger conclusions/takeaways are more buried (e.g., last paragraph of S2, last sentence of S5) throughout the text.


**Additional Feedback:**

This could be important work: my comments point to ways to clarify and help ML/CS readers take the full messages away from this paper. I believe this is achievable editing before a camera-ready and would make this paper much more impactful.

**Clarity:**

See the "weaknesses" discussion: technical ideas from both the ML literature and the fairness literature are introduced without context, which makes this more difficult for a broader audience to read. Some of the stronger conclusions/takeaways are more buried throughout the text. To successfully do this bridging work across fields, this paper needs to do more of the interpretation (show not tell).

- Section 5 contains some of the core arguments about values that are relevant for this audience; can any of these ideas be integrated earlier or combined with section 4? In addition the section on "fairness objectives" seems very important for this paper, but it is short and the last two sentences are not clearly written.
- Section 6 has good content but it is hard to see the takeaways, and it feels out of place. Can these ideas be combined elsewhere, moved into background, or even just swapped with section 7?

Lesser details:
- S3: problem with FTA as binary outcome - the last sentence seems out of place
- S3, last section: This first sentence is hard to parse and understand, and needs to be better connected to the second.
- S5: this section title is hard to parse. Can this be "Normative Values Embedded in Use of RAI Datasets" or similar?
- S5: this text is hard to parse, technical terms undefined, last two sentences difficult to read
- S6 has some spelled-out references rather than numeric.
Some unnecessary acronyms:
- DUA, STS, PSA (as well as typo in footnote 4); CS instead of spelling out or instead of AI/ML
- could drop acronyms from "educational and psychological research associations AERA, APA, and NCME"
- FTA should be introduced where it is being used regularly; right now it is introduced twice, and the first time out of useful context


**Documentation:**

N/A

**Ethics:**

No - ethics of other sources are the focus, and thinking tools are introduced for benchmark datasets

**Relation To Prior Work:**

Yes - integrates criminal justice, science and technology studies, psychology, and fair ML.

**Summary And Contributions:**

This paper engages with the use of risk assessment instrument (RAI) datasets as benchmarks in fair ML. The paper shows how the criminal justice field, psychology, law, and other fields have come together to create RAIs; what assumptions and values are embedded in the creation and use of RAIs; and how those embedded assumptions and values are unavoidable if RAI datasets are being used. The authors show how the typical framing of benchmark datasets hides these assumptions, which may lead to harmful outcomes---as well as statistically ill-informed results. Given the importance of the criminal justice context, the popularity of ("fair") ML, and the ubiquity of the COMPAS dataset, these authors provide important context and guidelines for the fair ML community.

---

### Author Response · Authors · 2021-07-13
**Response to Major Comments (Part 1)**

We thank all reviewers for their encouragement and thoughtful feedback.
We first jointly address major reviewer comments below (and will respond separately to each reviewer to clarify minor comments or nits):

We appreciate R1 and R3's recognition of the impact of our work on the ML community and the potential to magnify it by better integrating our discussion of technical fair ML results with clear insights on the takeaways and limitations of those results.

For section 3, we strongly agree with R1 that more clearly providing our interpretation of how well existing work addresses the problems we highlight, and how our analysis highlights remaining gaps in technical results, will be a strong contribution. We felt it was important to comprehensively acknowledge work in these areas but the current exposition reflects space limitations. We will use the extra camera-ready page to include the synthesis at the end of each subsection. Re R1's questions on how to interpret technical results in general, in summary, there is progress on studying these issues in isolation; in our overview, we try to specifically highlight the different informational requirements for proposed solutions. In the CJ setting, the problem is doubly so, in that unlike other settings we have much less ability to instantiate the informational requirements of proposed methodological solutions, and generally lack “ground-truth”. Lines 88-91, which point out the limited ability of purely technical solutions to address measurement error in CJ, try to do this pre-summary but we also plan to make this more explicit both in summary and in discussion of specific solution concepts.

For Section 4, R1 points out a lack of clarity in the intention and framing of the section. Our argument is on the limitations of algorithmic/mathematical definitions of fairness in translating to more fair outcomes in practice. We distinguish this from measurement error because this limitation refers to the difference between predicted outcomes and outcomes in practice, not the difference between measured and true values. We focus on points of judicial discretion as an example of how an RAI's downstream effects can be complicated and strongly dependent on the context in which they are deployed, and thus claims of more fair outcomes overlook how decision-makers influence outcomes. This argument is distinct from Section 3 since Section 4 discusses how downstream decision-making complicates predicted fairness gains, whereas Section 3 focuses on how inputs into models contain measurement errors. We plan to make the high-level argument, that algorithmic/mathematical fairness gains are limited in translating to more fair outcomes in practice, more explicit in the beginning of the section.

R1 suggests that ideas from Section 5 may be better incorporated into Section 4. However, we view Section 5 as very distinct from Section 4. Section 4 discusses how algorithmic/mathematical fairness definitions fail to capture how real-world decision makers use RAI outputs and thus the impact of RAIs on the real world, while Section 5 discusses ethical values of working with RAI datasets at a high level, such as supporting reform of the existing CJ institution. Even if algorithmic/mathematical fairness definitions could account for points of discretion mentioned in Section 4, Section 5's values of RAI/CJ research would still be present. We can make this distinction more explicit from the beginning.

Regarding the relevance of subsection "Fairness Objectives" in Section 5, we plan to use the additional page to cite existing work on the tradeoffs between fairness metrics (which is relatively well-studied in CS) and more explicitly tie it to ethical values. I have also included a rewrite of the last two sentences to be more clear: "Values motivating the choice between substantive and formal equality are perhaps related to how individuals perceive the causes behind differences in underlying distributions across protected attributes. For example, accounting for the historical context of systemic oppression, individuals may attribute that as a reason Black individuals have higher base rates of crime compared to white individuals by some metrics [77, 78], thus valuing the alleviation of social hierarchies instead of the reinforcement and substantive equality over formal equality."

We hope the above have addressed most of R1's large concerns, but will continue to address nit concerns or anything else not covered.

---

### Author Response · Authors · 2021-07-13
**Response to Major Comments (Part 2)**

Below is our response to the R3's larger concerns - we will address their nit concerns in a following response.
In short: R3 requested a table or specific datasheet for COMPAS; but our team lacks the institutional knowledge to speak to decisions made in construction of COMPAS. We instead propose to add a table highlighting COMPAS-specific issues known to the CJ community.

We thank R3 for their appreciation of the interdisciplinary domain contexts that we integrated into our paper, and are encouraged by their suggestion of a more structured empirical/methodological analysis around the issues with RAI datasets like COMPAS. We completely agree that this contribution would be incredibly valuable to the community, and think that a datasheet for COMPAS would be the best way to rigorously document the data distribution, the features, etc. to guide proper use in CS. This contribution is something we discussed as a group but ultimately decided not to do for a few reasons. A datasheet for COMPAS is a different contribution that requires institutional knowledge, thoughtfulness and deep insight into how the dataset was created, design decisions, etc., that we do not have access to as we did not collaborate with the original authors of COMPAS. Furthermore, many issues we discuss are not unique to COMPAS alone but rather applicable to many pretrial RAI datasets or RAI datasets in general, and we provide valuable insights for working with RAI/CJ data in general. And so, the problems go beyond just COMPAS and are shared structural characteristics of pretrial RAI datasets; to focus too much scrutiny on COMPAS could be misleading and give a wrong impression that another pretrial RAI dataset could be used as a benchmark in COMPAS’ place instead. That being said, we hope our work helps translate technical and data-level nuances relevant to fair ML research, to understand which specific data considerations are relevant to document in the first place.

To provide a framework for the future publication of a complete and rigorous datasheet for COMPAS, we have included examples of known issues with the COMPAS dataset that we will integrate into Section 3. We propose to add a table which illustrates how our framework can be applied to taxonomize COMPAS-specific issues: columns include a description of issues known to CJ community, translation of this issue to CS implications, and a line reference to where we discuss the general instantiation in our paper. Below we include a markdown table. (See endnotes for more detail).


| Issue with COMPAS known to CJ community | CS translation | Line ref. |
|---|---|---|
| Choice of threshold used for assessing classification disparities was arbitrary | Incorporate Discretion | lines 120-122, 206-213 |
| 2-year follow-up period includes arrests after case closed (no longer pretrial) | Bias in Y | 100 |

Regarding R3's concerns of scope of this paper, we believe that our paper falls under "identifying significant problems with existing datasets and their use" in the CFP rather than "frameworks for responsible dataset development" given we focus on pretrial RAI dataset use, not creation/development. Like R3, we also questioned whether our paper was in scope and reached out to the chairs of the track prior to submission, who expanded the scope of the track in response to our inquiry and reaffirmed that it is in scope.

Endnotes:
Elaboration on issues in table:
Issue 1, classification thresholds: Follow-up work that uses COMPAS (including the ProPublica analysis) translates continuous, probabilistic risk scores to classification discrepancies to aid disparity analysis. But even the highest-risk individuals have low rates of rearrest. For example only 26% of people with the highest risk scores on the Public Safety Assessment were re-arrested when validated in Kentucky. Classification discrepancies may not be the best way to assess disparity and papers that use COMPAS often assess arbitrary thresholds.
Issue 2, follow-up period: The ProPublica dataset followed everyone for two years even if that meant following them for a time after their case was closed. Once a person’s case closes, a new arrest that happens after that doesn’t count as an additional rearrest predicted by the initial RAI, it’s a new arrest for which a new (updated) RAI should be filled out and the prediction period starts over.

---

> ### Comment · Reviewer_4EMA · 2021-07-15
> **Response to major rebuttal**
>
> Thank you for acknowledging the need for a summary table for COMPAS in the paper. I do believe that this will make this paper very strong. I actually had not been expecting a complete detailed datasheet (describing the data collection etc) for the very reasons that you detailed, and I am satisfied with the table suggestions in your response.

---

### Author Response · Authors · 2021-07-15
**Response to Minor Comments (Part 1, R1+R3)**

# To Reviewer 1:

Thank you for the extensive suggestions and for articulating the key strengths of the paper. Regarding the weaknesses, we agree that we could make these edits for a camera-ready submission.
To summarize our response, some of these disconnects you point out arose from the space constraint, but an additional page in the camera-ready surely allows us to finalize by adding additional signposting and integration across sections; expositional polish rather than substantive changes to the paper or argument.

Clarifications on further minor points:
Re: “Some ideas are introduced without context..”:
- measurement error primarily refers to issues with operationalization.
- yes, we have adopted the typical use of “label bias” in ML which means "bias in the assignment of labels". We can certainly clarify the use of terms.
- we could add a sentence to emphasize “given predictive multiplicity of models, previous decisions could have been made on the grounds of these normative considerations without affecting accuracy”.
- we will clarify, the use of “distributional harms” is shorthand from the policy/economic context referring to finer-grained analysis of the distribution of benefits and harms under a policy decision (here, algorithm). For example, assessing conditional performance measures such as TPR measures distributional harm but other broader measures of harm are possible.
Thank you for the citations, we will add and discuss these.
Re: “the conclusion seems to be written …”: we propose to add an additional sentence summarizing the (negative) point directed at generic use of RAI datasets, “We have argued that using the COMPAS dataset as a generic benchmark, perhaps in deference to convention, is a poor choice for technical and normative reasons.”

# To Reviewer 3:

We thank you for your suggestions and address nits below.

You suggest adding citations for the sentence: "For example, it is not uncommon for the experiment design to be so disconnected from context that a prediction of high-risk is treated as the preferred outcome for an individual in an equal opportunity model simply because high-risk is the “positive” label in the dataset." We'd like to clarify that this is something we have seen as conference reviewers which then gets corrected before publication if the paper is accepted. So there is nothing to cite, but we feel it is a very illustrative anecdote. However, we can replace the phrase "not uncommon" to be more careful about the claims we’re making.

Re: rewriting the sentence on bail being a "positive" label for clarity, we include the rewrite here: It is also not uncommon in the CS community for the positive decision to be described as “granted bail.” (cite https://papers.nips.cc/paper/2017/file/82161242827b703e6acf9c726942a1e4-Paper.pdf)  However, granting bail is not equivalent to pre-trial release.  For the many presumptively innocent people who are incarcerated due to an inability to afford cash bail [citations], the decision to “grant bail” was, in practice, a decision to detain and is in no way “positive.”

Re: rewording/rearranging Section 6 and Section 8, we agree with your interpretation of the intention of Section 6 and we will change the title from "Standards and Guidelines" to "Established Norms" since not all of the content is guidelines for CS researchers. That being said, we also agree that we can make takeaways and insights for technical researchers more clear and explicit for the section. Here are a few takeaways that we will integrate into the corresponding subsections.
1. Data collection. AI/ML researchers should contact CJ researchers, particularly those responsible for data sets, to discuss decisions made during data collection/dataset creation rather than make assumptions about the data. Further, AI/ML researchers should consider collaborating with CJ researchers to obtain data and create datasets because AI/ML researchers have data organization/management skills CJ researchers likely do not, especially when dealing with large datasets.
2. Standards. These standards reflect decades of research in the area. AI/ML researchers should be cognizant of these standards because they will clarify some of the decisions CJ researchers have made and because this is largely the work that’s informing risk assessment in practice currently.
3. Guidelines. Using the data from RAIs without understanding how RAIs are actually used in practice leads to false conclusions. Further, new research is unlikely to see uptake in practice if it contradicts the way RAIs are actually being used and/or if it contradicts ethical or legal guidelines.
4. Language Guidelines. In the quest for fairness researchers can perpetuate unfairness if they use language that contributes to the marginalization of the very people for whom they are trying to create more fair algorithms

---

> ### Comment · Reviewer_4EMA · 2021-07-15
> **Response to minor rebuttal**
>
> Re: "not uncommon" -- I agree, although I am still wondering if the entire sentence can be rewritten in a way to make it clearer that this is an anecdote, or alternatively if you manage to find any example (even in non peer-reviewed papers, i.e., preprints) that would be useful too, although I realize that its a lot of work that you probably want to avoid given it is a nit.
>
> Thank you for agreeing with my comments otherwise. I am satisfied with the proposed changes, and I will increase my review score assuming these changes make it to the paper.

---

### Author Response · Authors · 2021-07-15
**Response to Minor Comments (Part 2, R2)**

# To Reviewer 2:

We thank you for your thoughtful feedback and are deeply encouraged that this paper resonated. We plan to add to the sentences mentioned that could profit from elaboration given the additional page in the page limit.  We will also go through the list of typos and again thank you for your close reading of the paper.  We were also interested in thinking about how simulated data could address these concerns - we will consider adding a part in Section 3 about this, or at the very least, suggesting it as a future area of research in the Call to Arms given the answer is, like most of these issues, complicated.

---

### Decision · Program_Chairs · 2021-07-26

**Decision:**

Accept

**Comment:**

This paper examines limitations and broader social consequences of the widespread use of risk assessment datasets -- specifically COMPAS-- within algorithmic fairness benchmarking. The authors demonstrate how these datasets embed values and assumptions that can lead to harmful outcomes, invalid results, and are poor methods of measuring real-world impact.

There is clear consensus amongst the reviewer that this paper is a strong contribution to this track. Reviewers note that the authors engage deeply with relevant literature, provide a tangible framework for understanding the structure of benchmarking problems, and provide clear guidelines for ML fairness practitioners.

Reviewers note that the paper is quite dense and the camera-ready draft could be improved (and made more accessible to a wider audience)  by elaborating on technical ideas from ML and algorithmic fairness literature as well as concepts that might not be familiar to the average ML practitioner, e.g. concepts relating to measurement modeling.